# Sox10-dependent neural crest origin of olfactory microvillous neurons in zebrafish

**Ankur Saxena, Brian N Peng, Marianne E Bronner***

Division of Biology, California Institute of Technology, Pasadena, United States

**Abstract** The sense of smell in vertebrates is detected by specialized sensory neurons derived from the peripheral nervous system. Classically, it has been presumed that the olfactory placode forms all olfactory sensory neurons. In contrast, we show that the cranial neural crest is the primary source of microvillous sensory neurons within the olfactory epithelium of zebrafish embryos. Using photoconversion-based fate mapping and live cell tracking coupled with laser ablation, we followed neural crest precursors as they migrated from the neural tube to the nasal cavity. A subset that coexpressed Sox10 protein and a neurogenin1 reporter ingressed into the olfactory epithelium and differentiated into microvillous sensory neurons. Timed loss-of-function analysis revealed a critical role for Sox10 in microvillous neurogenesis. Taken together, these findings directly demonstrate a heretofore unknown contribution of the cranial neural crest to olfactory sensory neurons in zebrafish and provide important insights into the assembly of the nascent olfactory system.

## Introduction

The last few decades have witnessed a surge of new information regarding the mature olfactory system, ranging from extensive and elegant descriptions of molecular and cellular functionality to physiological data on odorants, receptors, and transduction pathways. In contrast, far less is known about initial development of the olfactory epithelium during embryogenesis. This knowledge is critical for understanding not only disorders such as anosmia—the loss of smell—but also for clarifying the remarkable ability of the olfactory epithelium to regenerate neurons in adult organisms (*Graziadei and Graziadei, 1979*; *Bermingham-McDonogh and Reh, 2011*). In addition to providing critical clues for comprehending de novo differentiation of nascent stem cells into olfactory neurons, studies of olfactogenesis may offer mechanistic insights into neurogenesis as a whole.

The peripheral nervous system of the vertebrate head is derived from two sources: cranial ectodermal placodes and neural crest cells. Whereas placodes form the sense organs (nose, ears, and lens) of the head and contribute to neuronal portions of cranial ganglia, the glia are solely neural crest derived (*Northcutt and Gans, 1983*; *Baker and Bronner-Fraser, 2001*; *Barraud et al., 2010*). Sensory neurons are the primary functional players of the olfactory epithelium and have long been thought to be similar to neurons of the ear or eye. The latter arise within the developing sensory structures, the otic placode and neural retina, respectively. By analogy, the olfactory epithelium arises from the olfactory placode, a swathe of thickened ectoderm that initially resides within the anteriormost portion of the neural tube and subsequently moves laterally within the plane of the ectoderm. These cells invaginate to form the olfactory vesicle, which in turn becomes encapsulated by neural crest cells that form the surrounding nasal capsule (*Osumi-Yamashita et al., 1994*; *Le Douarin and Kalcheim, 1999*).

In the zebrafish embryo, the olfactory placode first becomes apparent around 18 hr post-fertilization (hpf) (*Hansen and Zeiske, 1993*; *Miyasaka et al., 2007*) and subsequently gives rise to sensory neurons and supporting cell populations. Within the olfactory epithelium, there are two predominant types of olfactory sensory neurons: ciliated and microvillous neurons. These differ in roles, receptors, location, and patterns of innervation. Whereas more basally located ciliated sensory neurons detect volatile odorants, more apically located microvillous sensory neurons can detect pheromones, nucleotides,

**\*For correspondence:** mbronner@caltech.edu

**Reviewing editor**: Robb Krumlauf, Stowers Institute for Medical Research, United States

**eLife digest** Neurons have crucial roles in both the peripheral and central nervous systems. The role of the neurons in the sensory organs (the eyes, ears, and nose) is to sense stimuli—including light, sound, and odor—and transmit this sensory information to the neurons of the central nervous system for processing. The first step in sensing an odor relies on the peripheral nerves of the olfactory epithelium. This tissue, which lines the inside of the nasal cavity, includes two main types of olfactory sensory neurons: ciliated sensory neurons that detect volatile or easily evaporated substances and microvillous sensory neurons that detect pheromones, nucleotides, and/or amino acids.

During vertebrate embryogenesis, an embryo develops three distinct germ layers, the ectoderm, mesoderm, and endoderm, each of which gives rise to the different tissues of the body. The ectoderm has three parts—the external ectoderm, the neural crest, and the neural tube—and together they give rise to the neurons of the peripheral and central nervous systems. The neurons within the eye and ear are known to originate from a thickened portion of the ectoderm, and it has been proposed that olfactory neurons develop in a similar manner.

Now, Saxena et al. show that, unlike what happens in the eye and ear, some olfactory sensory neurons originate from the neural crest. By studying the development of the olfactory system in zebrafish, Saxena et al. discovered that microvillous neurons, but not ciliated neurons, develop from neural crest cells, and that the transcription factor Sox10 is critical for the development of microvillous neurons. By establishing that neural crest cells are involved in the development of a substantial proportion of olfactory sensory neurons, this work sets the stage for future studies of olfactory nerve growth and regeneration. It may also assist researchers working on anosmia (the inability to smell).

and/or amino acids, depending on the species (*Buck, 2000*; *Sato and Suzuki, 2001*; *Hansen and Zielinski, 2005*).

The possible role of the neural crest in development of olfactory-related derivatives has been the subject of vigorous debate. Recent literature demonstrates that olfactory ensheathing glia, nonneuronal supporting cells previously assumed to derive from the olfactory epithelium, are actually neural crest derived (*Barraud et al., 2010*). These findings are consistent with the neural crest origin of all other peripheral glia (*Woodhoo and Sommer, 2008*). In neuronal lineages, there has been considerable controversy over the cell type of origin. As one example, some studies (*Whitlock et al., 2003*; *Forni et al., 2011*) have suggested that neural crest cells give rise to GnRH cells of the terminal nerve, which is close to though not part of the olfactory nerve and epithelium. However, many other studies refute this notion and support a placodal origin for GnRH cells (*Schwanzel-Fukuda and Pfaff, 1989*; *Wray et al., 1989*; *Dubois et al., 2002*; *Palevitch et al., 2007*; *Metz and Wray, 2010*; *Sabado et al., 2012*). Within the olfactory epithelium itself, *Harden et al. (2012)* recently examined the interactions of Sox10:eGFP-positive cells with six4b:mCherry-expressing placodal precursors in zebrafish. The authors interpreted their negative results to indicate a lack of contribution of neural crest cells to the olfactory epithelium. However, using the same Sox10:eGFP line, here we definitively demonstrate the neural crest origin of microvillous neurons within the olfactory epithelium.

To address the role of the neural crest in the development of olfactory sensory neurons, we have employed a combination of live confocal imaging, photoconvertible fate mapping, and laser ablation using zebrafish as a model. Interestingly, we find that neural crest gives rise to microvillous, but not ciliated, sensory neurons in the developing olfactory epithelium. Specifically, we demonstrate that a neurogenin1 (Ngn1) reporter and Sox10 protein are expressed in the subset of neural crest-derived nasal cavity cells that undergo dynamic ingression into the olfactory epithelium to form microvillous neurons. If the neural crest is ablated, the olfactory epithelium cannot compensate for the loss of microvillous neurons. Finally, we show that the neural crest specifier gene Sox10 is required for microvillous neurogenesis as these cells are ingressing into the olfactory epithelium. These results definitively demonstrate for the first time a Sox10-dependent neural crest origin for microvillous sensory neurons that take residence within the placode-derived olfactory epithelium.

## Results

### Live imaging of neural crest migration

To analyze the contribution of neural crest cells to the forming olfactory organ, we examined fluorescent neural crest cells in live transgenic zebrafish embryos at high resolution via confocal microscopy using a Sox10:eGFP line (*Wada et al., 2005*; *Carney et al., 2006*; see 'Materials and methods' for full annotation of all used lines). Visualization and analysis with Imaris software (Bitplane, Zurich, Switzerland) demonstrated that cranial neural crest cells moved anteriorly from the dorsal neural tube toward the future location of the olfactory pit. Some neural crest cells migrated and coalesced as a cup-like structure around the olfactory vesicle (*Figure 1A,B*) to form the mesenchymal nasal cavity, consistent with descriptions in mouse, chick, and medaka (*Langille and Hall, 1988*; *Osumi-Yamashita et al., 1994*; *Le Douarin and Kalcheim, 1999*).

Subsequently, beginning at ~24 hpf and continuing until at least day 2, a subset of the Sox10:eGFP-positive cells in the nasal cavity delaminated and intercalated into the olfactory epithelium where placode-derived ciliated neurons and support cells were already present. Concomitantly, these cells increased their eGFP levels and changed their morphology from mesenchymal to the 'teardrop' shape characteristic of neurons (*Figure 1C–J* and *Videos 1 and 2*).

To characterize the nature of the ingressing neurons, we crossed the Sox10:eGFP line with OMP:RFP and TRPC2:Venus lines, which bear markers of ciliated and microvillous neurons, respectively (*Sato et al., 2005*). The results show that all of the Sox10:eGFP-positive cells within the olfactory epithelium are microvillous neurons, as revealed by their TRPC2:Venus expression and apical position (*Figure 1K* and *Figure 1—figure supplement 1*). In contrast, none are OMP:RFP-positive ciliated neurons (data not shown). A small number of microvillous neurons are Sox10:eGFP-negative (*Figure 1K* and *Figure 1—figure Supplement 1*, arrowheads), raising the possibility that a minor subset of these cells has a separate origin. To further confirm that the ingressed cells are neurons, we performed immunostaining with an antibody against HuC/D, a marker for newly born post-mitotic neurons (*Kim et al., 1996*), which colocalized in the apically located Sox10:eGFP-positive neurons (*Figure 1—figure supplement 2*).

All neurogenic placodes in zebrafish, including the olfactory placode, express the proneural bHLH transcription factor Ngn1, which is known to be required for sensory neuron differentiation in mice (*Andermann et al., 2002*; *Bertrand et al., 2002*; *Cau et al., 2002*). In addition, Ngn1 is expressed in neural crest cells that become sensory neurons in the dorsal root ganglia in zebrafish (*McGraw et al., 2008*). Using a Ngn1:nRFP transgenic line (*Blader et al., 2003*) that has been well-characterized as representative of endogenous Ngn1 expression (*Blader et al., 2003, 2004*; *Madelaine et al., 2011*), we noted Ngn1:nRFP expression in a subset of Sox10:eGFP-positive cells in the nasal cavity (*Figure 1B*). Interestingly, only cells that upregulate Sox10:eGFP and coexpress Ngn1:nRFP ingress into the olfactory epithelium to differentiate into microvillous neurons (*Figure 1F*).

### Lineage tracing by photoconversion confirms a neural crest origin for microvillous neurons

Although Sox10-driven lines mark neural crest cells, Sox10 is not entirely neural crest specific and becomes downregulated with time in some lineages, whereas highly stable eGFP often persists. To substantiate our real-time confocal imaging, we followed the migration and differentiation of small groups of neural crest cells by making use of the PhOTO-N transgenic zebrafish line (*Dempsey et al., 2012*) expressing a nuclear localized version of the highly stable photoconvertible (green to red) protein Dendra2 (*Gurskaya et al., 2006*). By crossing this line to Sox10:eGFP fish (*Figure 2A*), we were able to photoconvert small numbers of neural crest cells (*Figure 2B,B'*), ensuring specificity of labeling. We then followed their movement from the neural rod to the nasal cavity. For some experiments, large blocks of cells (*Figure 2C,C'*) were photoconverted to achieve broad coverage and identify all possible neural crest contributions. In both cases, all photoconversion was performed between 14 hpf (10-somite stage) and 16 hpf (14-somite stage), always unilaterally.

Prior to day 1 of development, no photoconverted cells were observed in the olfactory epithelium (data not shown), consistent with our previous findings from the Sox10 reporter line on the timing of microvillous ingression (see *Figure 1*). At 29 hpf, a few photoconverted cells were present in the nasal cavity (*Figure 2D–E'*). At 55 hpf, by which time ingression/differentiation was nearly complete, photoconverted neural crest cells had differentiated into apically localized microvillous neurons. These results are consistent across embryos having either 'small' or 'large' numbers of photoconverted cells

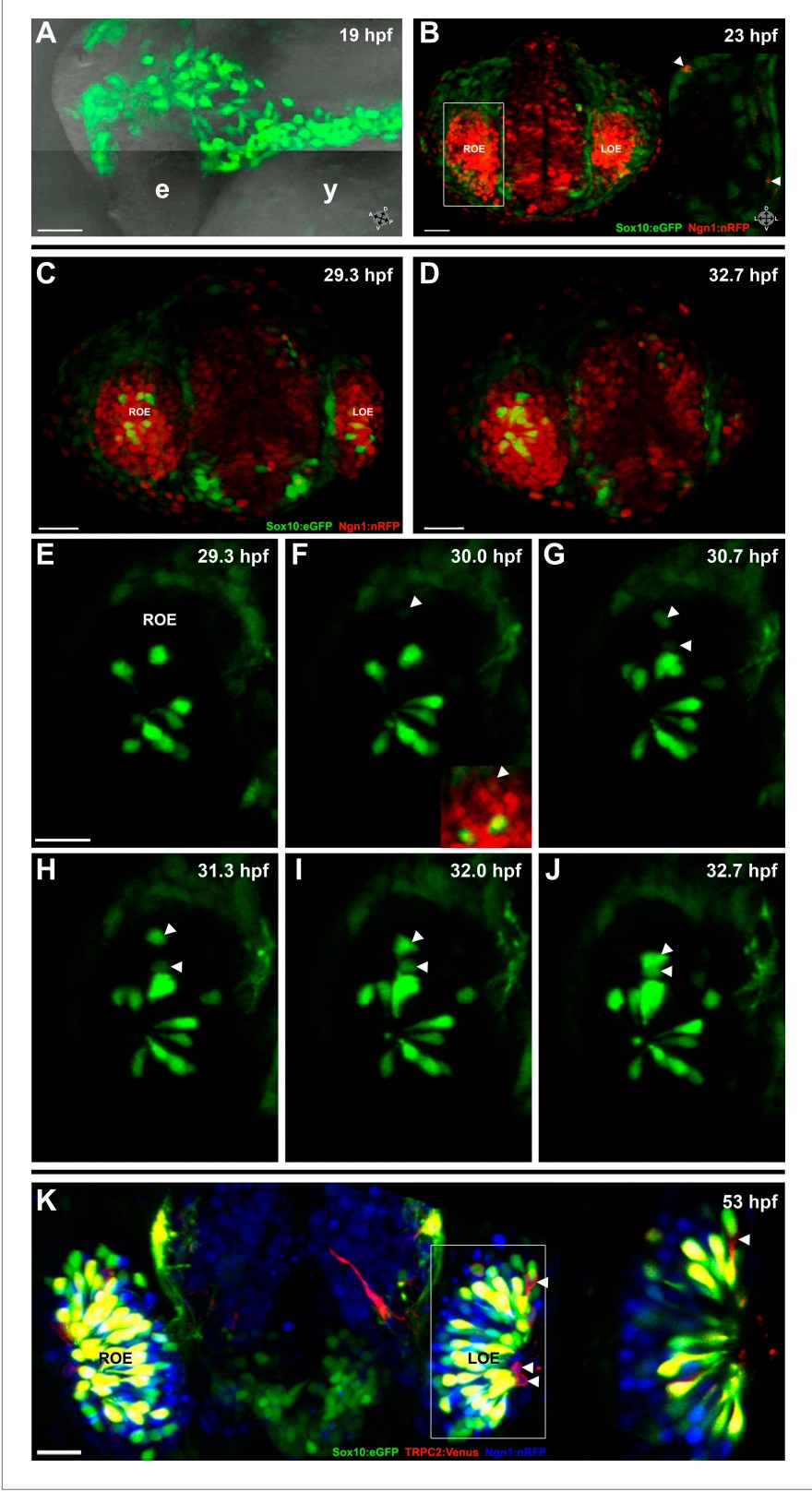

**Figure 1**. Sox10:eGFP+/Ngn1:nRFP+ cells migrate and differentiate into microvillous neurons. (**A**) Sox10:eGFP+
neural crest migrates dorsal to the eye and toward the olfactory region at 19 hpf as the olfactory placode is first
*Figure 1. Continued on next page*

*Figure 1. Continued*

apparent. (**B**)–(**D**) Time-lapse confocal microscopy of live embryos (Sox10:eGFP⁺; Ngn1:nRFP⁺) demonstrates that a subset of nasal cavity cells that express Ngn1:nRFP (**B**, boxed area arrowheads; z = 3.5 µm) ingress into the olfactory epithelium. (**E**)–(**J**) Selected time point and z-plane excerpts from full z stacks are shown from ~29 hpf (**C**) to ~33 hpf (**D**) and are 40′ apart; z = 17.5 µm (consisting of five 3.5 µm slices). Arrowheads indicate two ingressing cells, one from the top and the other directly behind the olfactory epithelium. All ingressing cells express Ngn1:nRFP (**F**, inset arrowhead). Sox10:eGFP, green; Ngn1:nRFP, red. (**K**) At 53 hpf, all Sox10:eGFP⁺ cells in the olfactory epithelium are TRPC2:Venus⁺ microvillous neurons. (because eGFP signal bleeds into the Venus channel, overlap was confirmed via image processing; see ***Figure 1—figure supplement 1***). Only a small number of microvillous neurons are not Sox10:eGFP⁺ (arrowheads). A single 3.5-µm-thick z-plane slice of the boxed area more clearly shows colocalization. Sox10:eGFP: green; TRPC2:Venus: red; Ngn1:nRFP: blue. e: eye; y: yolk; LOE: left olfactory epithelium; ROE: right olfactory epithelium. Orientation arrows: A: anterior; P: posterior; D: dorsal; V: ventral; L: lateral. Scale bars: 50 µm (**A**); 30 µm (**B**–**D**); 20 µm (**E**–**K**). See also ***Figure 1—figure supplements 1 and 2*** and ***Videos 1 and 2***.

The following figure supplements are available for figure 1:

**Figure supplement 1**. Panel (**K**) from ***Figure 1*** with Sox10:eGFP (green) channel removed (bottom) to better illustrate the large population of ciliated neurons present basally that are not Sox10:eGFP⁺ but are Ngn1:nRFP⁺ (blue).

**Figure supplement 2**. (A) and (A′) Sox10:eGFP⁺ microvillous neurons in fixed embryos stained with anti-GFP antibody colocalize with anti-HuC/D antibody staining at 53 hpf, confirming post-mitotic neuronal identity.

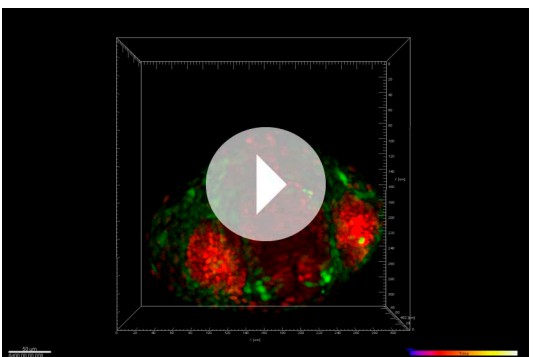

**Video 1**. Sox10:eGFP, Ngn1:nRFP from ~26.5 hpf to ~33 hpf, 20′ intervals. Over time, multiple Sox10:eGFP⁺, Ngn1:nRFP⁺ cells (bright green-yellow) appear in the apical portion of the olfactory epithelium and adopt neuronal morphology. See also ***Figure 1***. Videos assembled and registered using Imaris software.

(***Figure 2F–G′***). In contrast, no photoconverted cells differentiate into the basally located ciliated neurons, consistent with their placodal origin and confirming the specificity of our labeling. Interestingly, many labeled neural crest cells cross the midline and migrate to the contralateral side (shown in ***Figure 2***) where they differentiate into microvillous neurons. The small subset of microvillous neurons that are not Sox10:eGFP-positive show no overlap with the photoconverted Dendra2_Red (***Figure 2—figure supplement 1***).

## Laser ablation of Sox10:eGFP/Ngn1:nRFP-positive cells in the nasal cavity prevents microvillous neuron formation

Since our findings suggest that neural crest cells contribute the vast majority of microvillous neurons to the olfactory epithelium, we asked whether elimination of specific subpopulations of these cells would alter development within the olfactory epithelium and whether the olfactory epithelium could compensate for the loss of this source population.

Two-photon laser ablation was performed at 25 hpf unilaterally on small (2–7 cells, not shown) or large (≥20 cells) (***Figure 3A,B***) groups of neural crest-derived nasal cavity cells that were Sox10:eGFP positive and either Ngn1:nRFP negative or positive. These manipulations left cells in the olfactory epithelium untouched. By 35 hpf, both the small (***Figure 3C***) and large (***Figure 3D***) ablations of Ngn1:nRFP-positive cells in the right-side nasal cavity had highly deleterious effects on the subsequent number of microvillous neurons. By comparison, the unablated left-side nasal cavity and olfactory epithelium appeared normal. Large ablations often produced a striking phenotype with the near-complete absence of microvillous neurogenesis post-ablation (***Figure 3D***), whereas placode-derived ciliated neurons (Sox10:eGFP negative and Ngn1:nRFP positive in olfactory epithelium) continued to form. Ablations of Ngn1:nRFP-negative nasal cavity cells had a minor phenotype (data not shown), which may be due to the fact that some ablated cells were destined to turn on Ngn1 but were not yet expressing Ngn1:nRFP at detectable levels.

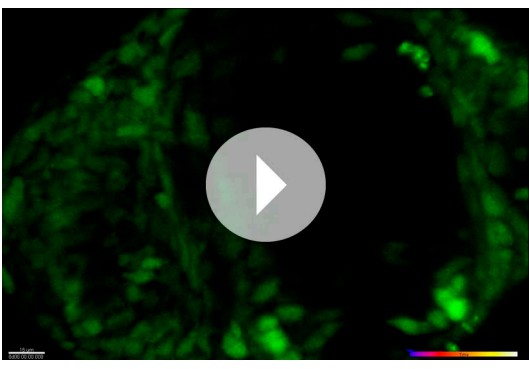

**Video 2**. Higher magnification view of the right olfactory epithelium, Sox10:eGFP channel only, taken from **Video 1**. Individual cell movements can now be clearly observed. Several Sox10:eGFP⁺ cells ingress from the nasal cavity into the olfactory epithelium and take on the teardrop shape characteristic of neurons while extending axonal projections toward the olfactory bulb. See also **Figure 1**.

Taken together, these results reveal an essential role for a specific subset of neural crest–derived cells in the nasal cavity that upregulate Ngn1 shortly before ingressing into the olfactory epithelium. Interestingly, the placode-derived olfactory epithelium is unable to compensate for the loss of the neural crest–derived population of microvillous neurons.

## Sox10 protein levels dynamically increase in differentiating microvillous neurons

The Sox10:eGFP line exhibits a striking upregulation of eGFP expression in cells concomitant with their intercalation into the olfactory epithelium (see **Figure 1**). To examine if the increased eGFP reflects an upregulation of endogenous Sox10 protein, we performed immunostaining with a Sox10 antibody (**Park et al., 2005**). These experiments reveal a marked increase in Sox10 protein levels in differentiating neural crest–derived microvillous neurons as they move from the nasal cavity (arrowhead) into the olfactory epithelium and transition from mesenchymal to the teardrop morphology characteristic of neurons at both 26 and 34 hpf (**Figure 4A–B′**). Directly adjacent ciliated neurons (unstained) do not upregulate Sox10.

## Sox10 is required for microvillous neuron formation

The rapid upregulation of Sox10 levels during microvillous neurogenesis raises the intriguing possibility that Sox10 protein may play a vital role in this process. To test this hypothesis, we performed loss-of-function analysis with a morpholino that has been well characterized as specifically blocking Sox10 protein production (**Dutton et al., 2001a**, **2001b**; **Prasad et al., 2011**; **Whitlock et al., 2005**). After morpholino injection, olfactory development was followed via real-time confocal imaging of transgenic lines.

The results show that loss of Sox10 leads to marked depletion of microvillous neurons in the olfactory epithelium (**Figure 5B,C′,E′,F′,H**) in comparison to control morpholino-treated embryos (**Figure 5A′,D′,G**). Neighboring ciliated neurons, in contrast, develop with only mild disruption (**Figure 5B,C,E,F**), suggesting that downregulation of Sox10 protein specifically affects formation of microvillous neurons. This phenotype was apparent as early as 29 hpf (**Figure 5A–C′**) and remained consistent at 33 hpf, 55 hpf, and beyond (**Figure 5D–H** and **Figure 5–figure supplement 1**). By 55 hpf, ciliated neurons were present in robust numbers comparable to those in the control embryos, while microvillous neurons remained scarce and those that did form appeared disorganized. Furthermore, the few remaining microvillous neurons were often TRPC2:Venus positive but Sox10:eGFP negative (**Figure 5—figure supplement 1**), suggesting that they were likely the small proportion of microvillous neurons not derived from the neural crest.

Since Sox10 is necessary for differentiation of many neural crest lineages, including the melanocyte lineage, its loss leads to a known 'colorless' phenotype (**Dutton et al., 2001a**, **2001b**). We were able to use this decrease in pigmentation as a surrogate marker for the degree of Sox10 knockdown. All assayed embryos (**Figure 5I**) were scored as 'high pigmentation' (**Figure 5J**) or 'low pigmentation' (**Figure 5K**) and then further separated into 'mild', 'moderate', or 'severe' morphological phenotypes. Surprisingly, microvillous neuron reduction is a consistent and common feature across all categories after knockdown of Sox10. These data suggest that microvillous neuron formation is exquisitely sensitive to reductions in Sox10 protein levels, whereas other Sox10-dependent processes such as pigmentation may be more resilient.

## Sox10 is required for ingression into the olfactory epithelium

The transcription factor Sox10 plays a variety of roles in neural crest development that differ in time and cell type. To determine if Sox10 is required specifically during microvillous neurogenesis, as

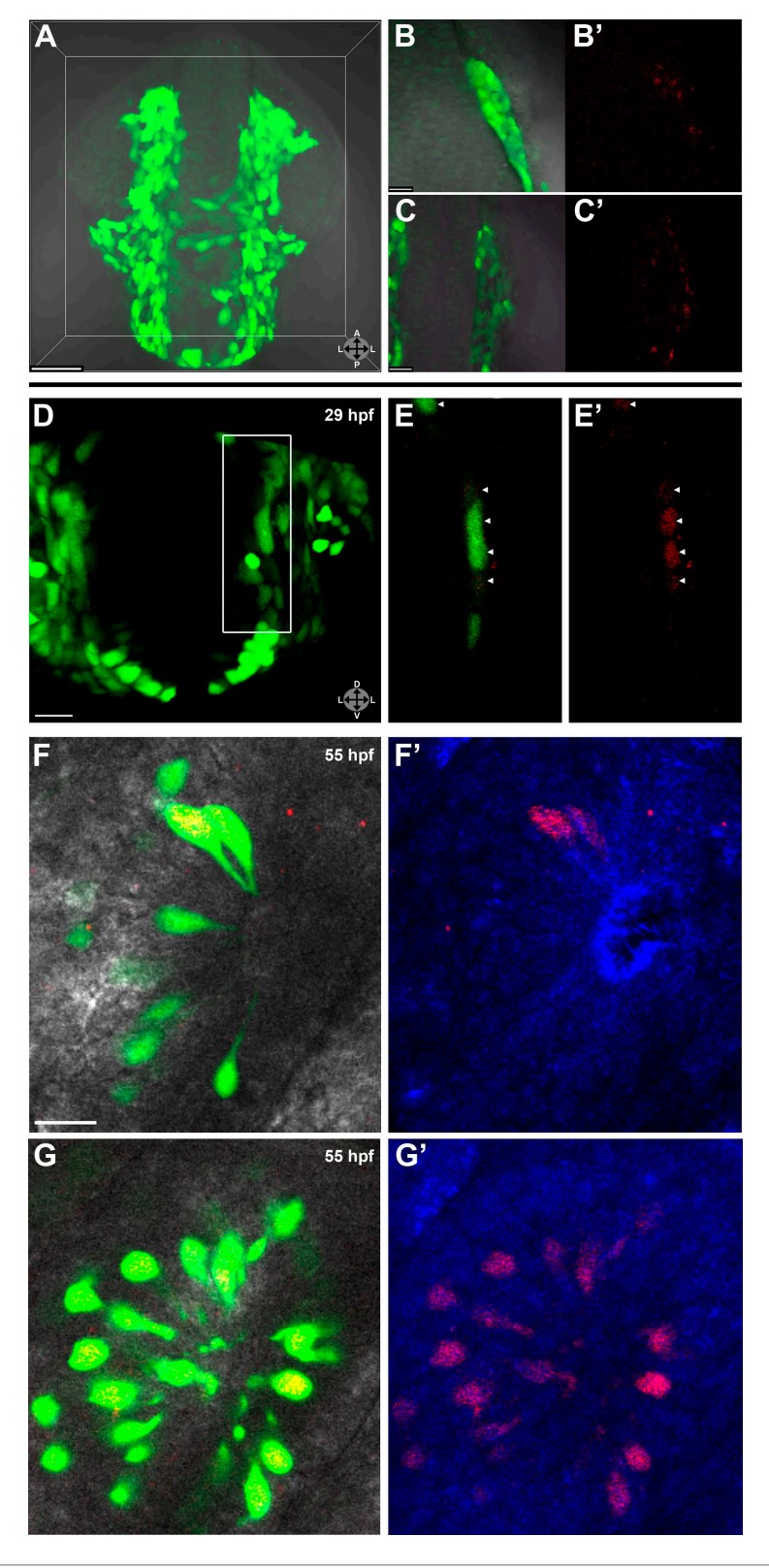

**Figure 2**. Lineage tracing by photoconversion demonstrates neural crest origin of microvillous neurons.
(**A**) Nuclear Dendra2 (low level green) was photoconverted in Sox10:eGFP+ (bright green) neural crest cells at
*Figure 2. Continued on next page*

*Figure 2. Continued*

14–16 hpf (10–14 somite stage) in live embryos. (**B**)–(**C'**) Representative example z-planes show unilateral photoconversion of a few cells (**B** and **B'**) or a large number of cells (**C** and **C'**). (**B** and **C**) show ubiquitous Dendra2 (low level green) and Sox10:eGFP (bright green), and (**B'** and **C'**) show photoconverted Dendra2 (red). (**D**) Photoconverted cells are visible in the contralateral nasal cavity at 29 hpf; (**E** and **E'**) show a 2.5-µm-thick z-plane slice of the boxed area with photoconverted cells (red, arrowheads). (**F** and **F'**) At 55 hpf, photoconversion of a small number of cells (**B** and **B'**) results in a subset of Sox10:eGFP⁺ microvillous neurons being labeled by photoconverted Dendra2 on the contralateral side. (**G** and **G'**) Large-scale photoconversion (**C** and **C'**) labels most Sox10:eGFP⁺ microvillous neurons. Directly adjacent ciliated neurons are never labeled. (**F**–**G'**) z = 2.5 µm; Sox10:eGFP: green; Dendra2$_{Green}$: blue; Dendra2$_{Red}$: red; histology in brightfield. Orientation arrows: A: anterior; P: posterior; D: dorsal; V: ventral; L: lateral. Scale bars: 50 µm (**A**); 20 µm (**B**–**D**); 10 µm (**F**–**G'**). See also *Figure 2—figure supplement 1*.
The following figure supplements are available for figure 2:

**Figure supplement 1**. Shown are a few 55 hpf microvillous neurons that are Sox10:eGFP negative (arrowheads) and were not photoconverted, suggesting a possible non-neural crest origin.

opposed to only during its well-established earlier role in neural crest specification/migration, we took advantage of a new photocleavable morpholino system (Gene Tools, LLC; *Tallafuss et al., 2012*). First, using a second translation-blocking antisense morpholino targeting Sox10 (previously used in *Dutton et al., 2001a*), we confirmed our previous results that Sox10 was necessary for microvillous neuron formation (data not shown). We then utilized a sense photo-morpholino specifically targeted to the second antisense morpholino and containing a photocleavable moiety. Coinjection of the two oligonucleotides sequestered the antisense Sox10 morpholino, rendering it inactive until illumination at 365 nm resulted in cleavage of the photo-morpholino and release of the antisense morpholino. This process allowed for the inhibition of Sox10 protein production from a specific time point onward and had the advantage of leaving Sox10-related processes such as early neural crest development intact. In addition, this experimental setup allowed for the injection of a single oligonucleotide cocktail into both control (not photocleaved) and experimental (photocleaved) embryos, reducing the variability inherent in morpholino-based experiments.

Injected embryos were subjected to photocleavage at the time of olfactory placode formation (~17.5 hpf) or at the onset of ingression into the olfactory epithelium (~24 hpf), with a subset of controls left unilluminated. In addition, some uninjected embryos were illuminated to control for any effects of UV light (data not shown). While control embryos developed normal numbers and organization of olfactory sensory neurons at 38 hpf (*Figure 6A,A'*) and 54 hpf (*Figure 6—figure supplement 1*), knockdown of Sox10 expression at 24 hpf specifically impeded microvillous neuron formation and organization (*Figure 6B,B'* and *Figure 6—figure supplement 1*). Knockdown at the earlier 17.5 hpf time point had an even more pronounced effect (data not shown), since it allowed more time for degradation of existing Sox10 protein before the start of ingression. Importantly, the later 24 hpf time point of photocleavage occurs well after neural crest cells have populated the nasal cavity, and results from that experiment decouple microvillous neurogenesis from earlier, critical roles of Sox10 in neural crest development and migration. Taken together, these results demonstrate that Sox10 is required for ingression/differentiation from the nasal cavity into the olfactory epithelium to generate microvillous neurons.

## Discussion

The data presented here reveal a previously unrecognized neural crest origin for one of the two main types of olfactory sensory neurons, microvillous sensory neurons, and implicate Sox10 as a critical player in microvillous neuron formation in zebrafish. Our conclusions are derived from several complementary lines of experimentation ranging from live imaging of migration to fate mapping via photoconversion and laser ablation to morpholino knockdown at specific time points. The cumulative results demonstrate that neural crest cells migrate anteriorly to form the mesenchymal nasal cavity. A subset of these cells expresses Ngn1 and upregulates Sox10 to high levels during ingression into the olfactory epithelium to form microvillous neurons. Laser ablation data confirm that the neural crest contribution is necessary for microvillous neurogenesis and that the olfactory placode is unable to compensate for the loss of this precursor population.

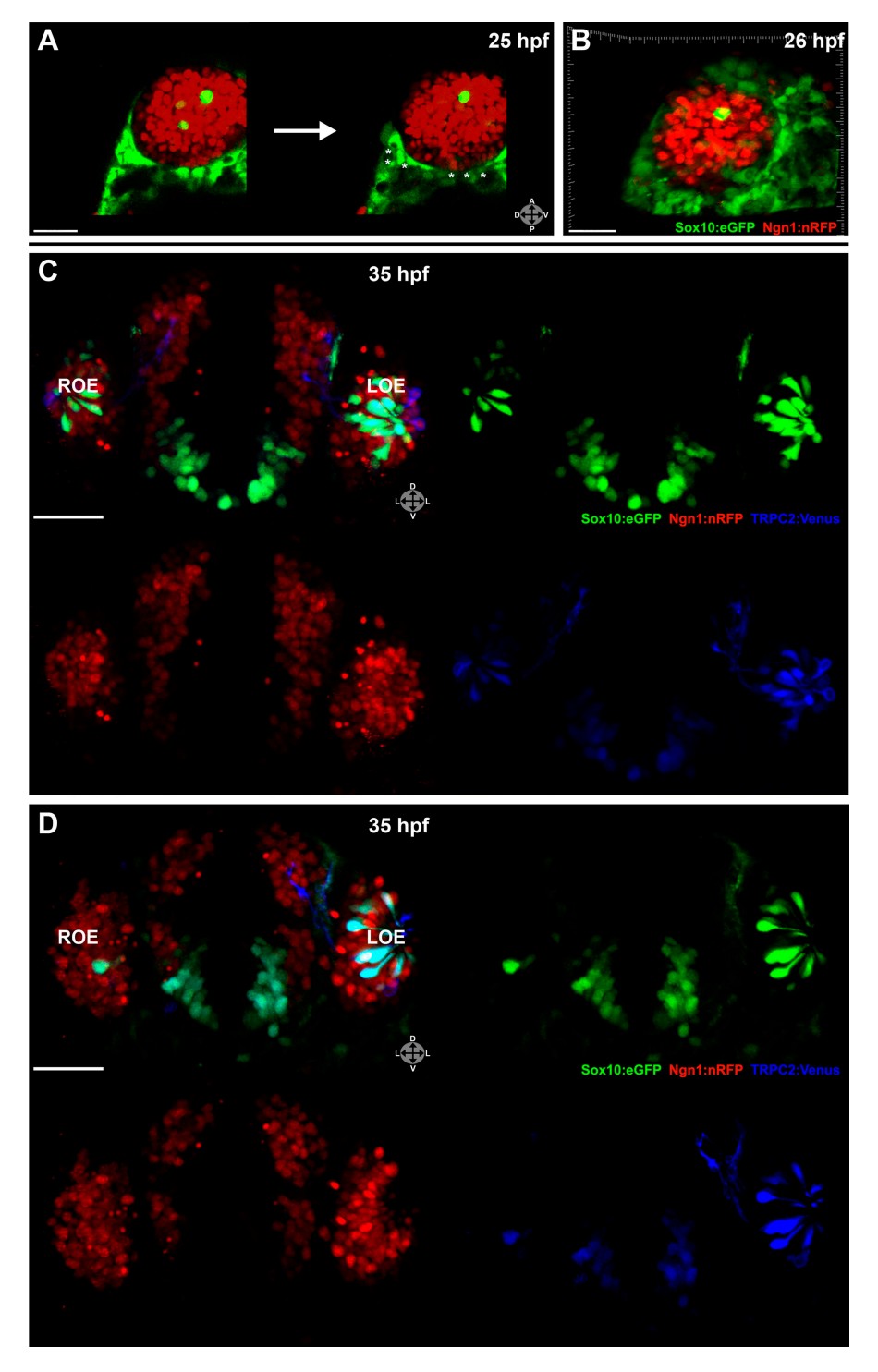

**Figure 3**. Laser ablation of neural crest in the nasal cavity inhibits microvillous neurogenesis. (**A**) Immediately before and after ablation of Sox10:eGFP+/Ngn1:nRFP+ cells in the right nasal cavity in live embryos; example z-plane slice shows six regions of ablation (asterisks). (**B**) 3D z-stack post-ablation of several z-planes with total of ≥20 Sox10:eGFP+/Ngn1:nRFP+ ablated cells. (**C**) and (**D**) 35 hpf embryos show a significant decrease in the number of Sox10:eGFP+/TRPC2:Venus+ microvillous neurons within the right olfactory epithelium as compared to the unablated left side in both small (**C**) and large (**D**) ablation experiments. Directly adjacent ciliated neurons are only marginally affected. Sox10:eGFP: green; Ngn1:nRFP: red; TRPC2:Venus: blue. LOE: left olfactory epithelium; ROE: right olfactory epithelium. Orientation arrows: A: anterior; P: posterior; D: dorsal; V: ventral; L: lateral. Scale bars: 30 μm (**A** and **B**); 40 μm (**C** and **D**).

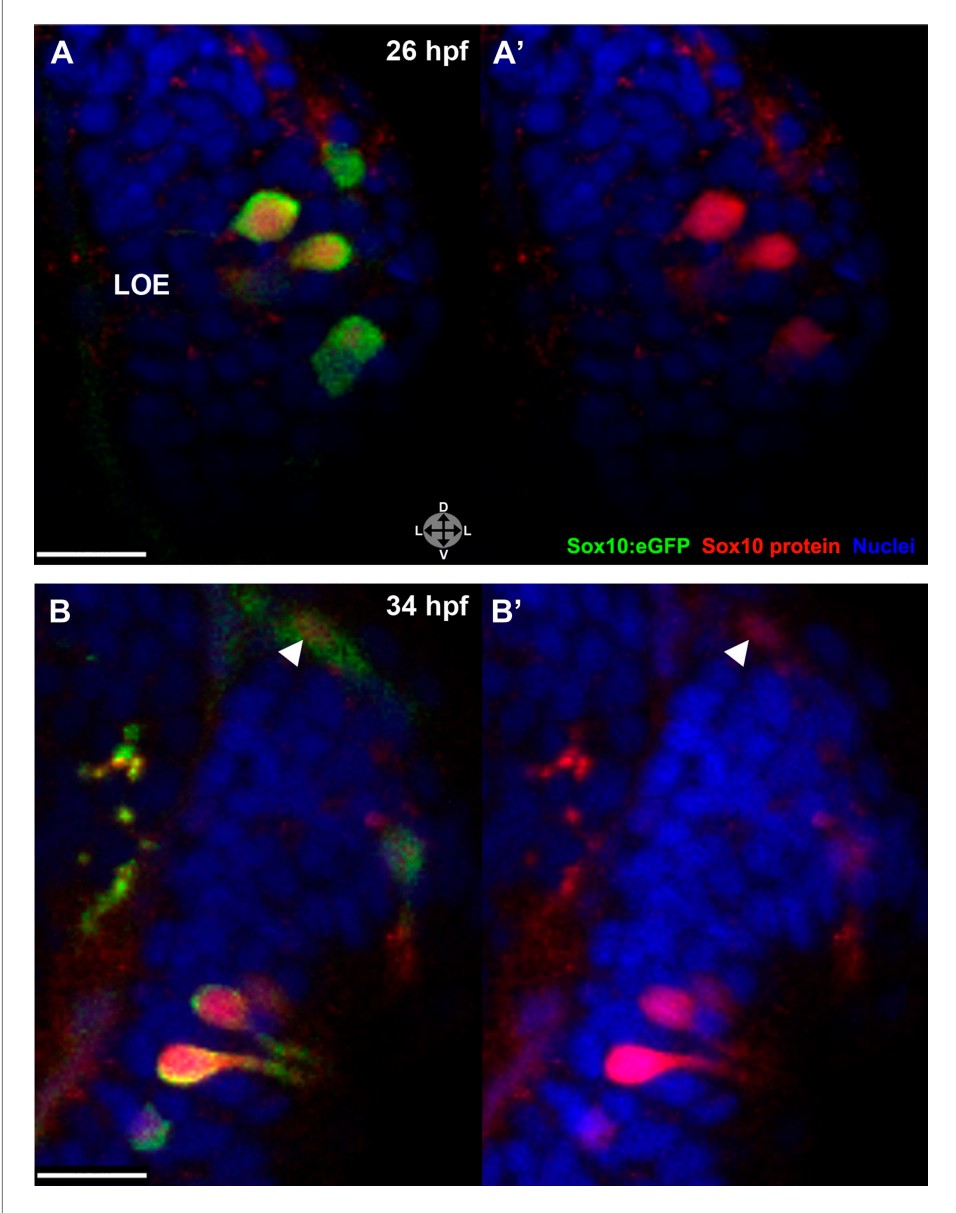

**Figure 4**. Sox10 protein is preferentially upregulated in neural crest cells that differentiate into microvillous neurons. (**A**)–(**B'**) Sox10:eGFP⁺ differentiating microvillous neurons in fixed embryos stained with anti-GFP antibody colocalize with anti-Sox10 antibody staining at 26 hpf (**A** and **A'**) and 34 hpf (**B** and **B'**). Levels of eGFP and Sox10 mirror each other, both increasing as cells ingress and become neurons. Arrowhead in (**B** and **B'**) indicates a nasal cavity cell likely about to ingress that has increased Sox10 protein expression. Sox10:eGFP: green; Sox10 protein: red; nuclear stain: blue; LOE: left olfactory epithelium. Orientation arrows: D: dorsal; V: ventral; L: lateral. z = 2.5 μm. Scale bars: 15 μm (**A**–**B**); 30 μm (**C**).

Our loss-of-function studies suggest that high levels of the transcription factor Sox10 are necessary for microvillous neurogenesis to proceed. Notably, the reduction in microvillous neurons after the loss of Sox10 is accompanied by only minor effects on the placode-derived ciliated neurons with which they intermix. Therefore, there is no general loss of neurogenesis in the olfactory placode but rather a microvillous-specific phenotype. An observed slight decrease in ciliated neuron numbers and thinning of projections in morpholino-treated embryos, similar to that seen in the laser ablation experiments, is likely a secondary effect due to the absence of normally adjacent microvillous neurons. Ciliated neurons largely recover by 48 hpf development, whereas neural crest–derived microvillous neurons do not.

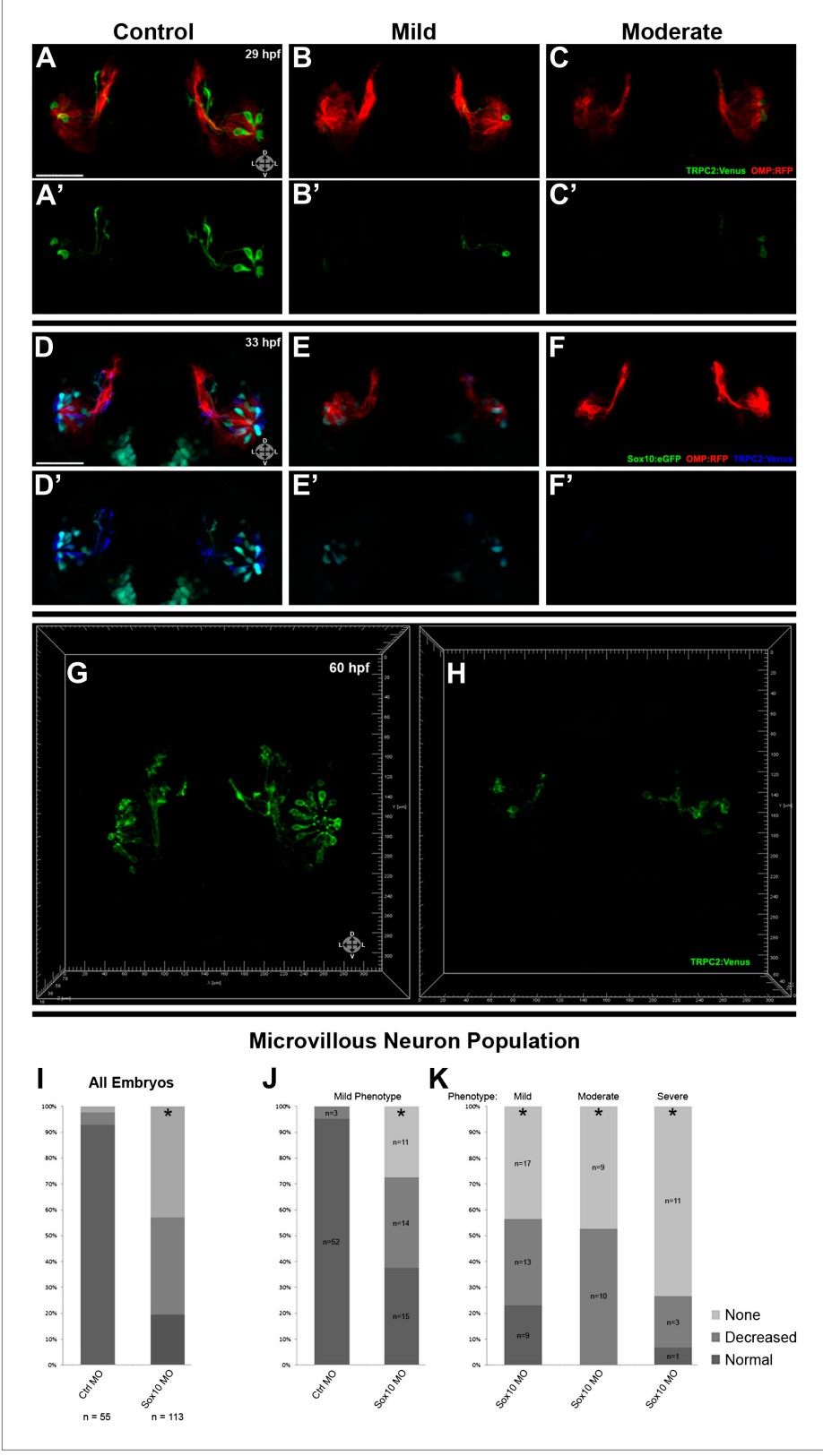

**Figure 5**. Morpholino knockdown of Sox10 selectively inhibits microvillous neurogenesis. (**A**) and (**A'**) Control morpholino-injected live embryos at 29 hpf have ciliated (red) and microvillous (green) neurons. (**B**)–(**C'**) Embryos
*Figure 5. Continued on next page*

*Figure 5. Continued*

with mild and moderate phenotypes (assayed histologically) display only slight changes in ciliated neurons, whereas microvillous neurons are significantly decreased in number and organization. (**D**)–(**F'**) Similar results are seen at 33 hpf, with almost no Sox10:eGFP$^+$ (green) cells becoming microvillous (blue) neurons. Ciliated (red) neurons remain relatively unaffected. (**G**) and (**H**) Antibody staining with anti-GFP against TRPC2:Venus in fixed embryos demonstrates a persistent decrease in the number of microvillous cells and disorganization at 60 hpf in Sox10 morpholino-treated embryos (**H**) in comparison to control embryos (**G**). All images were captured at identical settings to facilitate direct comparison of control and experimental embryos. (**I**)–(**K**) All embryos (**I**) were divided into 'high pigmentation' (**J**) or 'low pigmentation' (**K**) to roughly correlate Sox10 levels (higher and lower, respectively) with degree of olfactory phenotype. As expected, all control morpholino-treated embryos have high pigmentation, and all Sox10 morpholino-treated embryos with high pigmentation have only a mild phenotype. In all cases (**J** and **K**), Sox10 morpholino treatment results in most embryos having either no microvillous neurons or a decreased number. Ratios correlate with degrees of pigmentation and phenotype. (**A**–**C'**, **G**, and **H**) TRPC2:Venus: green; OMP:RFP: red. (**D**–**F'**) Sox10:eGFP: green; OMP:RFP: red; TRPC2:Venus: blue. Orientation arrows: D: dorsal; V: ventral; L: lateral. Scale bars: 50 μm. *p<0.001. See also ***Figure 5—figure supplement 1***.

The following figure supplements are available for figure 5:

**Figure supplement 1**. At 33 hpf, severe phenotype embryos (top right) have major malformations in both microvillous and ciliated neurons and often perish, likely reflecting non-specific effects. At 55 hpf (bottom) are shown the same embryos as in ***Figure 5 (D–F')***.

---

Interestingly, the role of Sox10 in microvillous neurogenesis is consistent with its role in regulating neuronal differentiation from neural crest of other peripheral sensory and autonomic neurons across species. Sox10 is not only required for formation of sensory, sympathetic, and enteric neurons but also for glia and Schwann cells (***Britsch et al., 2001***; ***Sonnenberg-Riethmacher et al., 2001***; ***Carney et al., 2006***; ***Kelsh, 2006***). It has been a matter of debate how much of this role is direct or indirect via Sox10's regulation of glial development (***Britsch et al., 2001***; ***Carney et al., 2006***). Data shown here, namely Sox10's expression in and specific effect on microvillous neurons within the first several hours of olfactogenesis, demonstrate a direct effect on neurogenesis at least in the context of zebrafish olfactory development.

The origins of various olfactory components have been the subject of considerable controversy in the literature. It was long assumed that olfactory sensory neurons derive solely from the olfactory placode (***Hansen and Zeiske, 1993***; ***Whitlock and Westerfield, 2000***). Although two lineage analysis studies in mice suggested that a small subpopulation of unspecified olfactory sensory neurons may have a neural crest origin (***Forni et al., 2011***; ***Katoh et al., 2011***), they relied upon Wnt1-Cre and P0-Cre, both of which suffer from lack of neural crest specificity. Furthermore, conflicting work by another group also using Wnt1-Cre lineage tracing found no contribution to neurons but rather a neural crest contribution only to olfactory ensheathing glia (***Barraud et al., 2010***). Interpretation of these experiments is made difficult by the fact that the Wnt1 marker is not entirely neural crest specific, as it is also expressed in the dorsal neural tube, otic placode, and several other regions (***Riccomagno et al., 2005***; ***Liu et al., 2006***; ***Freyer et al., 2011***; ***Valenta et al., 2011***). In zebrafish, ***Harden et al. (2012)*** suggest the idea of a possible neural crest contribution to olfactory ensheathing glia but not to sensory neurons, although they provide no direct evidence to substantiate this possibility. Here, by using photoconversion of Sox10-expressing neural crest cells, we were able to unequivocally follow individual neural crest cells from their site of origin in the dorsal neural tube to their ingression into the olfactory epithelium and differentiation into microvillous sensory neurons.

Olfactory sensory neurons are notable for their regenerative capacity. Hence, analysis of their lineage, paths of differentiation, and molecular determinants may provide important insights into general mechanisms of self-renewal. Since processes of renewal may be lineage-dependent, understanding the embryonic origins of different classes of olfactory neurons has distinct value. In the case of placode-derived ciliated sensory neurons, their replenishment throughout the lifespan likely occurs within the mature olfactory vesicle. Similarly, there may be a subpopulation of microvillous sensory neurons derived from the olfactory vesicle. Consistent with this possibility, we see a small number of microvillous neurons that are Sox10:eGFP-negative. Alternatively, given our evidence of large-scale neural crest contribution to microvillous sensory neurons, it is intriguing to consider that their replacement during adulthood may take place via a latent neural crest stem cell population derived from the nasal cavity.

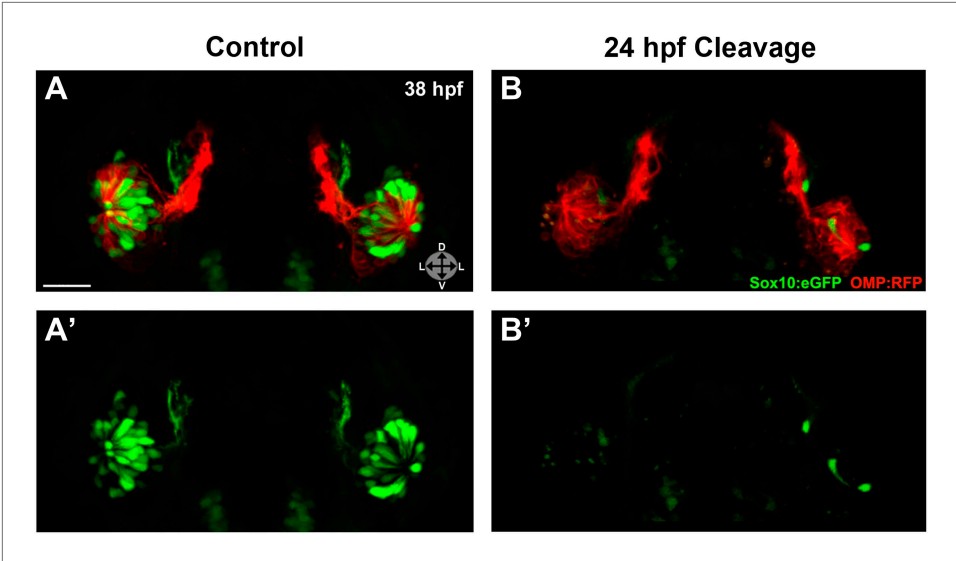

**Figure 6**. Photo-morpholino knockdown of Sox10 demonstrates its necessity during the ingression/differentiation process. (**A**) and (**A′**) Control (antisense morpholino + sense photo-morpholino injected but not photocleaved) live embryos have robust numbers of ciliated (red) and neural crest–derived microvillous (green) neurons at 38 hpf. In contrast, identically injected embryos subjected to photocleavage at 24 hpf go on to develop ciliated neurons but significantly lack microvillous neurons at 38 hpf (**B** and **B′**) in comparison to control embryos. Sox10:eGFP: green; OMP:RFP: red. Orientation arrows: D: dorsal; V: ventral; L: lateral. Scale bars: 30 μm. See also **Figure 6—figure supplement 1**.

The following figure supplements are available for figure 6:

**Figure supplement 1**. Shown are more details of the phenotype in **Figure 6**: (A and A′) control (antisense morpholino + sense photo-morpholino injected but not photocleaved) live embryos have robust numbers of ciliated (red) and neural crest–derived microvillous (green) neurons at 38 hpf. In contrast, identically injected embryos subjected to photocleavage at 24 hpf go on to develop ciliated neurons but have slight (B and B′) or, much more commonly, large (C and C′) decreases in microvillous neuron numbers and organization in comparison to control embryos.

In summary, we have utilized complementary techniques in zebrafish to directly demonstrate a neural crest contribution to olfactory neurogenesis that depends critically on the transcription factor Sox10. These data not only contradict existing dogma but open new avenues for better understanding sensory neurogenesis in the embryo and renewal in the adult. During vertebrate evolution, a myriad of changes have occurred in olfaction, including the emergence of the vomeronasal organ, which fish and birds appear to lack. Given the diverse nature of olfaction across species, it will be interesting to examine possible neural crest contributions from a comparative perspective as new lineage tracing tools become available in other model systems.

## Materials and methods

### Zebrafish

Wild-type and transgenic lines were maintained according to Institutional Animal Care and Use protocols. Embryos were grown, staged, and harvested as previously described (**Kimmel et al., 1995**; **Westerfield, 2000**). Treatment with 1-phenyl-2-thiourea (PTU) to prevent pigmentation from interfering with imaging was done for all experiments except morpholino knockdowns (to preserve observation of the 'colorless' phenotype). Lines used and their abbreviations are Tg(–4.9sox10:eGFP) (**Wada et al., 2005**; **Carney et al., 2006**) = Sox10:eGFP; Tg(-8.4neurog1:nRFP) (**Blader et al., 2003**) = Ngn1:nRFP; Tg(OMP2k:lyn-mRFP)/rw035 (**Sato et al., 2005**) = OMP:RFP; Tg(TRPC24.5k:gap-Venus)/rw037 (**Sato et al., 2005**) = TRPC2:Venus; Tg(bactin2:memb-Cerulean-2A-H2B-Dendra2)[pw1] (**Dempsey et al., 2012**) = PhOTO-N. Fish were mated in various combinations to yield compound

heterozygote embryos. Due to the spectral overlap between Venus and eGFP, fate identification crosses were done only with TRPC2:Venus confirmed homozygote adults.

## Live confocal imaging

Embryos were mounted in custom-designed embryo-shaped 1% agarose/30% Danieau molds to provide support while immersed in 2–3% methylcellulose/30% Danieau solution with standard working concentrations of tricaine anesthetic and PTU (to prevent pigmentation) added. Embryos were maintained at 28.5°C during imaging. Prior to and during experimentation, embryo staging was carefully monitored. A Zeiss LSM 510 Meta laser scanning confocal microscope was used with an LD C-Apochromat 40x/1.1 W Corr objective. Collected time-lapse data were registered using Imaris software (Bitplane) and exported as TIF images and AVI videos.

## Photoconversion

Sox10:eGFP (and sometimes also TRPC2:Venus) fish were crossed with PhOTO-N fish. Resultant dual- or triple-positive embryos were mounted as above for live imaging. Seven to ten embryos from a single clutch were photoconverted unilaterally in a D-V or near-lateral orientation per individual experiment, in four independent experiments. Photoconversion was performed as previously described (*Dempsey et al., 2012*), with the following modifications: a Zeiss LSM 5 Exciter with an LD C-Apochromat 40x/1.1 W Corr objective and Zen 2009 software's 'Regions' tool were used for photoconversion at 405 nm (200 cycles at 30% laser power) and immediate confirmation of conversion. Subsequent live imaging was done on a Zeiss LSM 510 Meta with an LD C-Apochromat 40x/1.1 W Corr objective of unconverted monomeric Dendra2$_{Green}$ excited at 458 nm and converted Dendra2$_{Red}$ excited at 561 nm. Each step (before conversion, conversion of selected cells, and after conversion) was captured as a single plane image, and full z-stacks were captured at the beginning and end for each photoconverted embryo. Embryos were imaged at several time points over multiple days to track the progress of photoconverted cells.

## Laser ablation

Sox10:eGFP[+]/Ngn1:nRFP[+]/TRPC2:Venus[+] embryos were mounted in a near-lateral angled orientation to allow ablation of olfactory epithelium cells (data not shown) or the nasal cavity surrounding the olfactory epithelium. There were 8–10 embryos per experiment, with two experiments directed at olfactory epithelium ablation and four at nasal cavity ablation, always unilaterally. Laser ablation was done with a two-photon laser at 840 nm for 20 cycles at 100% power on a Zeiss LSM 510 Meta with an LD C-Apochromat 40x/1.1 W Corr objective and Zen 2009 software's 'Regions' tool for cell selection. Subsequently, embryos were imaged to determine degree of ablation and overall health of both targeted cells and surrounding untargeted cells and were assayed with nuclear markers in some cases. Each step (before ablation, ablation of selected cells, and after ablation) was captured as a single plane image, and full z-stacks were captured at the beginning and end for each experimental embryo. Embryos were then imaged at several time points over multiple days to ascertain the post-ablation phenotype.

## Immunohistochemistry

Embryos were collected at specific time points, fixed with 4% PFA, and washed in PBST. They were then washed in PBDT (1% BSA, 1% DMSO, 0.1% Triton X-100 in PBS, pH 7.4), blocked in 10% normal donkey or goat serum/PBDT, and incubated overnight at 4°C with primary antibodies to HuC/D (1:100; Invitrogen # A-21271), GFP (1:250; Abcam # 6673), and/or zSox10 (*Park et al., 2005*) (1:500). Further PBST washes and blocking were followed by secondary antibodies (Invitrogen) overnight at 4°C. Hoechst 34580 was added to stain nuclei. After further PBDT and PBS washes, embryos were mounted for confocal imaging on a Zeiss LSM 5 Exciter with a Zeiss LD C-Apochromat 40x/1.1 W Corr objective. Each experiment was done at least in triplicate.

## Morpholino knockdown

All morpholinos were obtained from Gene Tools, LLC. A previously used translation-blocking antisense morpholino against Sox10 (ATGCTGTGCTCCTCCGCCGACATCG; *Dutton et al., 2001a*, *2001b*; *Prasad et al., 2011*; *Whitlock et al., 2005*) targets the −23 to +2 region of the Sox10 sequence. Based on previously used amounts in these papers, several concentrations in the 5–15 ng per embryo range were tested. All data reported here were obtained from injections of 7 ng (in 2.3 nl) of the Sox10

morpholino or control morpholino into one- to two-cell embryos. On day 1, embryos were sorted based on health and any aberrations (mild, moderate, and severe), the number of melanocytes (high and low), and the expression profile of Sox10:eGFP and/or TRPC2:Venus (normal, decreased/abnormal, and none). Multiple representative embryos were imaged on a Zeiss LSM 510 Meta with an LD C-Apochromat 40x/1.1 W Corr objective. Some embryos were fixed and stained with antibodies against GFP/Venus and/or Sox10 as outlined above. Numbers of control and experimental embryos are indicated in *Figure 5*.

### Photo-morpholino knockdown

A second previously used translation-blocking antisense morpholino against Sox10 (GCCACAGGTGACTTCGGTAGGTTTA; *Dutton et al., 2001a*) targets the −43 to −19 region of the Sox10 sequence. A sense photo-morpholino (AACCTACCGAPGTCACCTGTG) targets the antisense morpholino, with 'P' at position 11 indicating a photocleavable moiety. The antisense morpholino was used at 7 ng per embryo as described above for the first antisense morpholino. A premixed solution was made at room temperature of antisense morpholino and sense photo-morpholino at a 1:1.1 molar ratio and injected into one- to two-cell embryos. After testing several permutations, photocleavage described here was done at 17.5 or 24 hpf on a Zeiss Axiovert 35 fluorescent microscope with a 5× objective and illumination via an Opti-Quip Model 1200 mercury lamp paired with a G365/FT395/LP420 filter set for 15 min, three to four embryos at a time. Some control embryos were injected but not photocleaved, and other control embryos were uninjected but illuminated under the same conditions as injected embryos to control for any effects of UV light illumination. Embryos were then incubated until later time points, and those appearing healthiest were selected for imaging with a laser scanning confocal microscope as before. Three independent experiments were performed with 12–15 embryos imaged per experiment.

### Statistical analysis

Calculations to determine statistical significance of Sox10 morpholino experiments were performed using Student's t-test of variables (two-sample t-test assuming unequal variances).

## Acknowledgements

We thank Dr. Bill Dempsey, Dr. Periklis Pantazis, and Dr. Scott Fraser for sharing the PhOTO-N line pre-publication; LeighAnn Fletcher and David Mayorga for zebrafish husbandry assistance; Dr. Robert Kelsh for the Sox10:eGFP line; Dr. Uwe Strahle for the Ngn1:nRFP line; Dr. Yoshihiro Yoshihara, RIKEN BSI, and National Bioresource Project of Japan for the OMP:RFP and TRPC2:Venus lines; Dr. Bruce Appel for zSox10 antibody; Dr. Andres Collazo for laser ablation expertise; Dr. Scott Fraser and Abigail Saxena for feedback on the manuscript.

## Additional information

### Competing interests

MEB: Reviewing editor, *eLife*. The other authors declare that no competing interests exist.

### Funding

| Funder | Grant reference number | Author |
| --- | --- | --- |
| National Institutes of Health | R01 DE16459, HG004071 | Marianne E Bronner |
| Gordon Ross Postdoctoral Fellowship | | Ankur Saxena |
| National Institutes of Health | 5T32NS007251 | Ankur Saxena |

The funders had no role in study design, data collection and interpretation, or the decision to submit the work for publication.

### Author contributions

AS, Conception and design, Acquisition of data, Analysis and interpretation of data, Drafting or revising the article; BNP, Acquisition of data, Analysis and interpretation of data; MEB, Conception and design, Drafting or revising the article

## Ethics

Animal experimentation: This study was performed in strict accordance with the recommendations in the Guide for the Care and Use of Laboratory Animals of the National Institutes of Health. All of the animals were handled according to approved Institutional Animal Care and Use Committee (IACUC) protocol #1346 (Analysis of Neural Crest Lineage and Migration) at California Institute of Technology.

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
