## [Decision Letter]

Thank you for choosing to send your work entitled “Sox10-dependent neural crest origin for olfactory microvillous neurons” for consideration at *eLife*. Your article has been favorably evaluated by a Senior editor and 3 reviewers, one of whom is a member of our Board of Reviewing Editors.

The Reviewing editor (Robb Krumlauf) and the other reviewers discussed their comments before we reached this decision, and the Reviewing editor has assembled the following comments to help you prepare a revision.

This paper reports that the majority of microvillous neurons developing in the zebrafish olfactory placode derive not from placodal cells but rather from Sox10-positive neural crest cells that migrate to the placodal region. This conclusion is based on a combination of live time-lapse filming, fate mapping Sox10-labelled cells (both generically and targeted in a number of ways), and laser ablation of Sox10 neural crest cells. In addition, the authors show that Sox10 levels increase during development and that knockdown of this transcription factor either by conventional morpholino or by a clever photo-activatable morpholino (to control the timing) leads to loss of olfactory microvillous cells. Overall the study is very well done, nicely illustrated, and the authors' arguments are generally well supported.

1) There is a concern over the interpretation that Sox10 has a predominant effect during the ingression/differentiation process. It may also be required for early differentiation – it's just the later photo-activated morpholinos would not reveal that.

2) One common issue is whether it is possible that zebrafish olfactory development differs substantially from “dry-land” vertebrates. First, previous studies on the origin of GnRH neurons in zebrafish from Whitlock's group have provided contrary results to those in other species (apart from Sarah Bray's study in mice which agrees with the zebrafish studies, but there they rely on a genetic marker that may not be entirely specific). In this regard it is important that the authors cite and comment upon the following paper, which describes the migration of the sox10:GFP positive cells in the zebrafish: Harden et al, Developmental Dynamics (2012) 241:1143-1154. This shows that the sox10 GFP cells do not move into the placode but surround the placode, and the authors suggest that the sox10:GFP cells are/will become glia.

3) While there has been some controversy about the origin of the OE neurons, recent data (Barraud et al) has shown that in chicken and mouse olfactory ensheathing glia originally thought to be placodal in origin are in fact neural crest. Hence, data has pointed towards a role for neural crest in contributing glia and neurons to the OE. This current paper convincingly locks down the neural crest origins of the specific microvillous population in zebrafish. Because studies have been done with multiple experimental systems it would be important to see a coherent set of experiments in one system and/or address the possibility of differences between species. That there may be a difference in the olfactory system of different vertebrate classes might make sense if there is a marked difference in the function of this system in air and water. For example, are the olfactory ensheathing cells in zebrafish also of neural crest origin? The authors might have that information in their data set and it would greatly enhance the value of the story to have it in the paper. It would go a long way towards confirming the lineage relationships that they see are consistent with mouse and chick results, which they imply in the paper. Ideally the authors should test at least some of their observations in a “dry” vertebrate (mouse or chick), but if this is not possible, the paper should be made very explicitly relevant to zebrafish in the title and abstract, as well as in the paper, to accommodate this possibility and to avoid confusing readers.

---

## [Author Response]

*1) There is a concern over the interpretation that Sox10 has a predominant effect during the ingression/differentiation process. It may also be required for early differentiation – it's just the later photo-activated morpholinos would not reveal that*.

We agree with the reviewers that we were previously unclear in our statements and we have now clarified this important point. We now state that Sox10 likely plays multiple roles in neural crest development. By using photoactivable morpholinos, we show that it also is critical for olfactory microvillous neurogenesis during ingression/differentiation. However, we agree with the reviewers that Sox10 plays many roles in the neural crest and is clearly used reiteratively in several phases of neural crest development and many lineages.

*2) One common issue is whether it is possible that zebrafish olfactory development differs substantially from “dry-land” vertebrates. First, previous studies on the origin of GnRH neurons in zebrafish from Whitlock's group have provided contrary results to those in other species (apart from Sarah Bray's study in mice which agrees with the zebrafish studies, but there they rely on a genetic marker that may not be entirely specific). In this regard it is important that the authors cite and comment upon the following paper, which describes the migration of the sox10:GFP positive cells in the zebrafish: Harden et al, Developmental Dynamics (2012) 241:1143-1154. This shows that the sox10 GFP cells do not move into the placode but surround the placode, and the authors suggest that the sox10:GFP cells are/will become glia*.

The reviewers have raised excellent points, and we have now added discussion of this paper including a comparison to our own results. There are certainly important anatomical differences between fish and mammals, as mammals have a vomeronasal organ whereas fish do not, and the story in birds is ambiguous. Even within these structures there may be neuronal heterogeneity, which makes it critical to study these events in multiple species to increase understanding of the lineage relationships. Currently, mammalian experiments are severely limited by the lack of neural crest-specific labeling and the difficulty of live cell tracking. As new imaging technology becomes available in the future, we hope to overcome these technical limitations and gain new insights into these questions.

*3) Are the olfactory ensheathing cells in zebrafish also of neural crest origin? The authors might have that information in their data set and it would greatly enhance the value of the story to have it in the paper. It would go a long way towards confirming the lineage relationships that they see are consistent with mouse and chick results, which they imply in the paper. Ideally the authors should test at least some of their observations in a “dry” vertebrate (mouse or chick), but if this is not possible, the paper should be made very explicitly relevant to zebrafish in the title and abstract, as well as in the paper, to accommodate this possibility and to avoid confusing readers*.

This is a very interesting question that we wish we could address but found ourselves limited by currently available tools. The olfactory ensheathing cells (OECs) differentiate much later than neurons.

Unfortunately, the turnover of photoconverted Dendra2 limits the number of days that cells can be tracked. For this reason, we were unable to follow our labeled cells for long enough to observe cell bodies along the olfactory nerve. Further complicating this analysis, there are few/no good markers for glial cells in zebrafish. GFAP, a general glial marker in zebrafish, has a transgenic line described as having faint expression in some (unspecified, not shown) cells of the olfactory placode (Bernardos & Raymond, 2006), and a previously used anti-GFAP antibody in our hands is inconclusive or negative in olfactory labeling. Therefore, we are limited by the reagents available to test this interesting possibility. We are in the process of developing new transgenic tools that may allow this question to be answered, but the timeline precludes inclusion in this manuscript. As suggested, we have altered our title and abstract to make it clear that our data speak to sensory neurons specifically in the zebrafish model system. In addition, we have now included an additional neuronal marker (HuC/D) to further identify Sox10:GFP+ olfactory neurons.